# Acetate and glycerol are not uniquely suited for the evolution of cross-feeding in *E. coli*

**Magdalena San Roman**[1,2], **Andreas Wagner**[1,2,3]*

**1** Department of Evolutionary Biology and Environmental Studies, University of Zurich, Zurich, Switzerland, **2** Swiss Institute of Bioinformatics, Lausanne, Switzerland, **3** The Santa Fe Institute, Santa Fe, New Mexico, Washington, United States of America

* andreas.wagner@ieu.uzh.ch

**Data Availability Statement:** All relevant data are within the manuscript and its Supporting Information files.

**Funding:** AW acknowledges support by ERC Advanced Grant 739874, by Swiss National

## Abstract

The evolution of cross-feeding among individuals of the same species can help generate genetic and phenotypic diversity even in completely homogeneous environments. Cross-feeding *Escherichia coli* strains, where one strain feeds on a carbon source excreted by another strain, rapidly emerge during experimental evolution in a chemically minimal environment containing glucose as the sole carbon source. Genome-scale metabolic modeling predicts that cross-feeding of 58 carbon sources can emerge in the same environment, but only cross-feeding of acetate and glycerol has been experimentally observed. Here we use metabolic modeling to ask whether acetate and glycerol cross-feeding are especially likely to evolve, perhaps because they require less metabolic change, and thus perhaps also less genetic change than other cross-feeding interactions. However, this is not the case. The minimally required metabolic changes required for acetate and glycerol cross feeding affect dozens of chemical reactions, multiple biochemical pathways, as well as multiple operons or regulons. The complexity of these changes is consistent with experimental observations, where cross-feeding strains harbor multiple mutations. The required metabolic changes are also no less complex than those observed for multiple other of the 56 cross feeding interactions we study. We discuss possible reasons why only two cross-feeding interactions have been discovered during experimental evolution and argue that multiple new cross-feeding interactions may await discovery.

## Author summary

The evolution of cross-feeding interactions, where one organism thrives by consuming the excretions of others, can create diversity even in simple and homogeneous environments. In past work, we had predicted that 58 cross-feeding interactions could evolve in populations of *E. coli* grown in glucose minimal media, yet only two have been experimentally observed, those involving acetate and glycerol. We hypothesized that multiple mutations might be required for the evolution of computationally predicted but not experimentally observed cross-feeding interactions. To answer this question, we developed a method that searches for the minimal number of metabolic changes required for individuals to change their metabolic state (from an ancestral glucose-consuming state to

Science Foundation grant 31003A_172887, as well as by the University Priority Research Program in Evolutionary Biology at the University of Zurich. The funders had no role in study design, data collection and analysis, decision to publish, or preparation of the manuscript.

**Competing interests:** The authors have declared that no competing interests exist.

an evolved state that produces or consumes other metabolite). We observed that the metabolic changes required for the evolution of acetate and glycerol cross-feeding are no less complex than those required for the evolution of the other predicted cross-feeding interactions, which suggests that multiple cross-feeding interactions may still await discovery.

## Introduction

One trillion microbial species have been predicted to inhabit our planet [1]. To understand how life on earth became so enormously diverse is a central goal of ecology and evolutionary biology. For many decades, most biological diversity was thought to arise when populations become physically subdivided [2], allowing mutations to accumulate independently in each subpopulation. More recently, biologists have increasingly accepted that populations can also diversify without any physical barrier [3–7] when organisms specialize and adapt to different niches available in a heterogeneous environment [8,9]. For instance, when apples were introduced to North America, some apple maggot flies changed their plant host from hawthorn to apple. Today, apple maggot fly populations feed on hawthorns or apples. Such emerging ecological barriers can lead to the creation of new species.

Remarkably, diversity can also evolve in homogeneous environments [6,7,10–12]. Perhaps the most striking example involves stable genetic polymorphisms that originated in populations of *Escherichia coli* cultured in homogeneous batch or chemostat environments. Initially isogenic populations of an ancestral *E. coli* strain developed genetic polymorphisms which coexisted over hundreds of generations in eleven out of fifteen evolution experiments performed in a chemostat [10]–a culturing device in which a cell culture is kept in a constant environment with the continuous addition of fresh medium and the removal of culture liquid containing leftover nutrients, metabolic waste products, and microbial cells. The chemical environment used in these experiments was a minimal medium containing glucose as the only carbon source. Diverse strains isolated after approximately 800 generations from one of these parallel experiments showed that glucose-acetate and glucose-glycerol cross-feeding enabled the coexistence of these strains. That is, one strain consumed the primary carbon source present in the medium (glucose) and excreted a secondary carbon source (acetate or glycerol), whereas the other strains fed on the excreted secondary carbon source. Genome sequencing of the ancestral and evolved strains [13] revealed almost 600 mutations in the evolved strains. Approximately 30 repeatedly mutated genes encode enzymes involved in glucose uptake, central metabolism, fermentative pathways, the TCA cycle, glyoxylate shunt and phospholipid biosynthesis. The ancestral strain itself harbored further regulatory mutations in genes required for acetate and glycerol catabolism, which may have predisposed it to evolve acetate and glycerol cross-feeding interactions [13,14]. The polymorphisms that evolved in the other parallel experiments have not been so thoroughly analyzed, but it has been shown that acetate cross-feeding was also responsible for the maintenance of five other polymorphism [15].

Similar cross-feeding emerged when the experiment was performed in batch culture, an environment different from a chemostat, where nutrients get depleted, waste products accumulate, and cell densities rise over time, before a sample of the culture is transferred into fresh medium. In this experiment, an isogenic population of *E. coli* cultured in a minimal glucose medium also diversified into coexisting strains which persisted for at least 10000 generations in nine out of twelve populations [12]. Genome sequencing of two such strains in one of these populations showed that they emerged after 6500 generations and coexisted due to glucose-acetate cross-feeding [11]. Their emergence was possibly facilitated by the population's hypermutator phenotype–the clones harbored on average 199 mutations [16].

Experiments like these suggest that *E. coli* readily diversifies genetically and metabolically in a completely homogeneous environment by filling niches that do not exist in the environment but are created by the organism itself. We are interested in finding out how much microbial diversity can be created through such cross-feeding by characterizing the whole spectrum of molecules (beyond acetate and glycerol) that can be cross-fed. In recent work we discovered through metabolic modeling that all metabolic systems have a large potential for the evolution of cross-feeding interactions. For example, we found that when *E. coli* feeds on glucose, 58 metabolites can be excreted as by-products of metabolism. Each of these metabolites can in turn serve as a carbon source that can help sustain a stable community of cross-feeding strains. Among these metabolites are acetate and glycerol, for which cross-feeding was observed experimentally. In other words, metabolic modeling predicts 56 additional cross-feeding interactions, which raises the question why none of these interactions have been observed experimentally. Possibly, many other such polymorphisms have indeed evolved but went undetected, because currently no systematic experimental screen for cross-feeding interactions exists. (The cross-feeding strains were detected through colony morphologies on agar plates, and substantial biochemical and genetic work was needed to prove that their polymorphisms resulted from cross-feeding.) A second possible reason is that not all predicted cross-feeding polymorphisms can evolve with the same likelihood. For example, metabolites may differ in the number of metabolic changes or DNA mutations needed to turn a strain into a producer or consumer of the metabolite. Here we explore this second possibility. That is, we ask whether glucose-acetate and glucose-glycerol cross-feeding have been observed because they are much more likely to evolve than other cross-feeding interactions.

Ideally, to quantify the likelihood that a given cross-feeding interactions evolves, it would be necessary to know all mutations that give rise to the evolution of a producer or consumer strain, the probability that each mutation takes place, and the mutation's fitness effect in every genetic background and in the population in which it occurs. This amount of information is not within reach of current technology, especially because the genetic changes leading to cross-feeding may be complex and involve changes in metabolic enzymes, regulatory molecules, and transport proteins [13,16]. The problem is aggravated by the fact that the same phenotypic change, such as the emergence of cross-feeding, can often be achieved through multiple and perhaps myriad different combinations of genotypic changes [17–19].

Faced with these obstacles, we here take a phenomenological approach, in which we estimate the likelihood for a cross-feeding interaction to emerge through the amount of metabolic change that an ancestral strain must experience to bring forth a producer and consumer strain for the cross-fed metabolite. In other words, we use an assumption of parsimony: The producer and consumer strains most likely to evolve from an ancestor are those that are metabolically most similar to the ancestor.

More specifically, we perform the following analysis for each of 58 metabolites that can form the basis of a stable cross-feeding polymorphism in a chemostat inoculated with a single ancestral *E. coli* strain and supplied with a glucose-minimal medium. We analyze genome scale metabolic model of *E. coli* (iJO1366, [20]) to identify the distribution of metabolic fluxes–the rates at which enzymatic reactions proceed–that are most similar between a producer strain of the metabolite and the ancestor, as well as between the consumer strain of this metabolite and the ancestor. We use multiple measures of similarity, among them the number of reactions that require a change in flux, and we assume that a producer or consumer strain is more likely to emerge if this number is small. Our analysis shows that acetate and glycerol cross-feeding do not require exceptionally small metabolic changes compared to the 56 other metabolites we consider.

## Results

### Complex metabolic changes are needed for the evolution of glucose-acetate and glucose-glycerol cross-feeding

The four stably co-existing and cross-feeding *E. coli* strains that evolved from an ancestral strain in a glucose-limited chemostat showed genetic and physiological differences with respect to glucose, acetate, and glycerol uptake and metabolism [10]. Our first analysis prepares the ground by modeling the ancestral and evolved flux distributions of *E. coli* that are relevant to reconstruct the lab-evolved glucose-acetate and glucose-glycerol cross-feeding interactions. We began by modeling the metabolic behavior of the ancestral strain using a modified version of Flux Balance Analysis (FBA) known as parsimonious FBA (pFBA), together with the *i*JO1366 genome scale metabolic network of *E. coli* (see Methods).

Flux balance analysis is a computational method for predicting metabolic fluxes for all reactions in a genome-scale metabolic network. Essentially, FBA identifies a flux distribution that results in maximal biomass growth while fulfilling a set of constraints. These include the assumption that metabolism operates in a steady-state, where the production and consumption of each metabolite are exactly balanced. The constraints also include assumptions about the reversibility of reactions, as well as about maximally possible rates of nutrient transport. Generally, there is multiple alternative flux distributions that fulfill all constrains and permit maximal growth. Among them, *parsimonious* FBA [21] identifies the flux distribution that minimizes the sum of all metabolic fluxes, which can be viewed as a proxy for the total expression level of metabolic enzymes. In other words, pFBA assumes that a metabolism must achieve maximal growth subject to minimal cost. This cost minimization increases consistency between computational predictions and transcriptomic and proteomic data [21].

We applied pFBA to determine the flux distribution of the ancestral strain from which cross-feeding emerged under the conditions in which its evolution had been observed experimentally [10]. (Further below, we determine the ancestral flux distribution with an alternative method and show that our observations are robust to the method used to find the ancestral flux distribution.) Specifically we assumed a chemically minimal environment where sufficient glucose–the sole carbon source–is available to allow growth at the dilution rate of the chemostat in the experiments that inspired this work (0.2 h$^{-1}$) [10]. (See methods for details.) Fig 1B illustrates part of the resulting flux distribution graphically for central carbon metabolism.

After having obtained this ancestral flux distribution we predicted the flux distributions of the evolved cross-feeding strains. We used the same metabolic network of *E. coli* (*i*JO1366) to model ancestral and cross-feeding strain. This modeling decision reflects the observation that cross-feeding strains can emerge in little evolutionary time [10,11], and that metabolic differences between strains are not due to differences in their enzyme-coding genes, but result from mutations that affect the expression of these genes or the activity of the encoded enzymes. In addition, we assumed that evolution is most likely to bring forth producer and consumer strains whose flux distribution differs as little as possible from the ancestor.

We first focused on the origin of the strain that produces acetate. Experimentally, this strain was found to consume more glucose than the ancestor, and to excrete acetate and glycerol into the environment. To disentangle the metabolic changes that are required for acetate and glycerol production, we modeled two separate producer strains: an acetate producer and a glycerol producer. To model the acetate producer strain, we imposed non-zero (1 mmol gDW$^{-1}$ h$^{-1}$) acetate excretion on this strain and allowed the strain to consume more glucose than the ancestral strain, because it could otherwise not persist in the chemostat. Specifically, we set glucose consumption to the minimal amount required to satisfy the acetate excretion constraint and to allow growth at the dilution rate value.

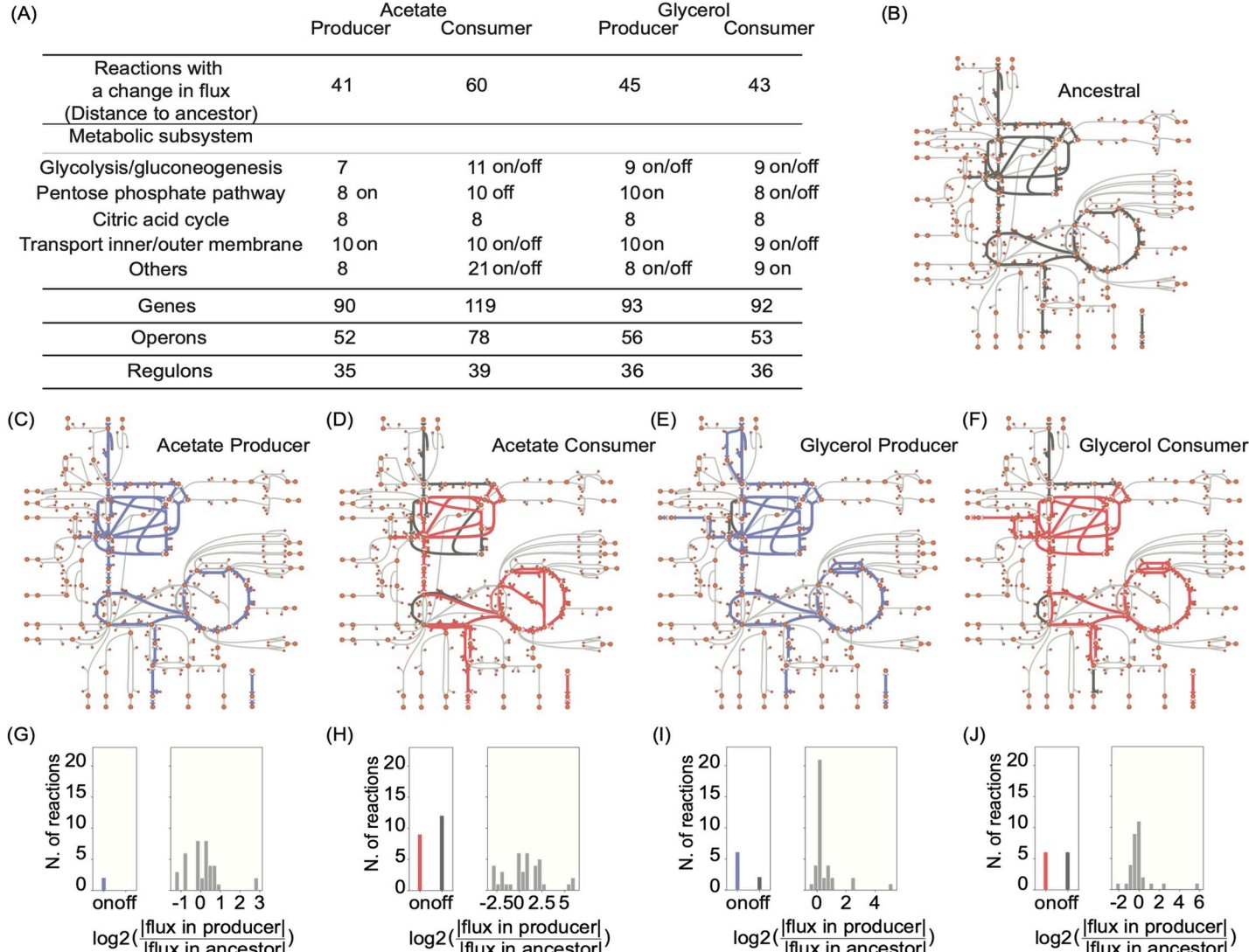

**Fig 1. Metabolic changes required for the evolution of the acetate and glycerol cross-feeding strains.** (A) Minimal number of reactions requiring a change in flux in the ancestor for acetate and glycerol cross-feeding to evolve. Most reactions requiring a flux change belong to glycolysis and gluconeogenesis, the pentose phosphate pathway, the citric acid cycle or from transport processes. The remaining reactions ('others') come from multiple pathways that comprise oxidative phosphorylation, alternate carbon metabolism, pyruvate metabolism, glycine and serine metabolism, alanine and aspartate metabolism, folate metabolism, anaplerotic reactions, or that are unassigned to a pathway. In the table we also include the number of genes associated with the reactions requiring a flux change, the number of operons into which these genes fall, as well as the number of regulons. (B) Central carbon metabolism of *E. coli*. Every orange circle represents a metabolite and every line a reaction. Thick grey lines indicate a non-zero flux in the ancestral strain ($|a_i| > 0.001$ mmolgDW$^{-1}$ h$^{-1}$). (C) to (F) As in (B), but each panel shows reactions with non-zero flux in cross-feeding strains (blue for producers, red for consumers) in addition to non-zero fluxes in the ancestor (grey). (S1 Fig allows 'zooming in' to see metabolite names, reaction names, and flux values.) Panels (G) to (J) show, on the left of each panel, the number of reactions that are activated in a producer (blue) or consumer (red) relative to the ancestor ('on'), or the reactions that are inactivated relative to the ancestor ('off', grey). On the right of each panel, the amount of flux change is shown for reactions that change their flux relative to the ancestor.

To reflect our parsimony (minimal metabolic change) assumption, we assumed that acetate production in this strain is achieved via the smallest possible flux rearrangement relative to the ancestral strain. More specifically, we used Regulatory On/Off Minimization (RooM), an optimization method that finds the flux distribution which minimizes the number of reactions whose flux needs to change from the ancestral distribution such that the strain can produce acetate.

We modeled the evolution of the second cross-feeding strain, the acetate consumer, analogously. Experimentally, this strain was found to consume glucose but distinguished itself from the ancestor and the other evolved strains by its large acetate consumption capacity. We modeled the acetate consumer strain by disallowing glucose consumption completely and permitting only acetate consumption, because it allows us to identify the metabolic changes that are associated with a change in carbon source most clearly. (We also repeated the analysis allowing the acetate consumer strain to consume both acetate and glucose, which led to the same conclusions. See S2 Fig)

Again, we applied RooM to identify the smallest number of reactions whose flux needs to change to bring forth an acetate consumer strain. We then repeated this entire procedure for both the glycerol producer and consumer. Fig 1C–1F show the flux through central carbon metabolism in the producers (in blue) and consumers (in red) on top of the flux distribution identified for the ancestral strain (in grey). We found that acetate (glycerol) production and consumption requires changes in fluxes through at least 41 (45) and 60 (43) reactions (Fig 1A and S1 Text). In other words, the required flux change, even though it is the minimally necessary change, is complex.

This complexity is also evident in different kinds of predicted flux changes. Some reactions are active (non-zero flux) in the ancestor but inactive ('off') in the evolved strain. Other reactions are active ('on') only in the evolved strain. Yet other reactions change only their flux magnitude in the evolved strain. Fig 1G–1J show the numbers of reactions in these three categories. Although the majority of reactions (65–95%) change only their flux magnitude, between 2 and 21 reactions need to be turned on or off to allow the production or consumption of acetate and glycerol.

In addition to comprising different kinds of changes, the changed fluxes fall into various metabolic subsystems, including glycolysis and gluconeogenesis, the pentose phosphate pathway, the citric acid cycle, and transport (Fig 1A). Thus, it is unlikely that they could be brought forth by one or few mutations, an assertion that is corroborated by our next analysis.

To find out how flux changes might be related to genetic changes, we used the Gene-Protein-Reaction association (GPR) map available for the metabolic model of *E. coli* iJO1366. The GPR map is only one-to-one in the simplest case, where a gene product catalyzes one reaction. Alternatively, a gene product may catalyze more than one reaction; the products of multiple genes may be needed to catalyze one reaction; or the products of different genes may catalyze the same reaction. The 41 reactions requiring a flux change to bring forth acetate producer strain are linked to 90 genes, which fall into 52 operons and 35 regulons. Fig 1A shows the number of genes, operons and regulons associated with the reactions requiring a flux change for the evolution of the acetate consumer and glycerol cross-feeding strains. The changes involve multiple operons and regulons. Consistent with the prediction that one or few mutations could not bring about all these changes, experimentally evolved cross-feeding strains harbored hundreds of mutations. Multiple repeatedly mutated genes [13] were involved in glycolysis and gluconeogenesis, the TCA cycle, and transport, which are three of the subsystems where we also observe most of the reactions changes. Only a limited number of mutated genes [13] directly map onto metabolic reactions that we predict to require a flux change (S3 Text and S1 File), and transcriptomic changes [13,14] also show very limited agreement with computational predictions (see S3 Text, S3 Fig and S2 File). This is unsurprising, because of the ambiguity of gene-reaction associations, and because gene expression change poorly reflects metabolic flux change for several reasons, for example because mRNA and enzyme abundance correlate poorly [22–27].

## Acetate and glycerol cross-feeding does not require exceptionally little metabolic change

A previous analysis of the metabolic network of *E. coli* showed that 56 distinct cross-feeding interactions other than the experimentally described glucose-acetate and glucose-glycerol interactions can evolve and lead to stable polymorphic communities [28]. (Note that our work only considers cross-fed metabolites that can either diffuse through the cell membrane or that can be excreted or imported by an *E. coli* transport protein.) This raises the question why only the latter two types of cross-feeding polymorphisms have been described. One possible answer is that the evolution of acetate and/or glycerol cross-feeding requires fewer metabolic changes than cross-feeding involving other metabolites, and is thus more likely to evolve. To find out, we repeated the analysis from the previous section, but replaced the secondary carbon sources acetate and glycerol with each of the other 56 metabolites that could potentially be cross-fed.

The minimal metabolic distances between ancestor and producer, as well as between ancestor and consumer, as obtained with RooM, are shown in Fig 2A. The mean producer-ancestor distance equaled 57±15 reactions with changed flux. The corresponding consumer-ancestor distance equaled 62±17 reactions. In 41 out of the 58 cross-fed metabolites (those situated above the diagonal in Fig 2A), the producer of a given metabolite can evolve more easily than its consumer, requiring fewer flux changes. The correlation observed between ancestor-producer and ancestor-consumer distances can be explained by the large overlap of reactions requiring a flux change during the evolution of producer and consumer strains, even though the magnitude and direction of the change differs between producer and consumer (S1 Text).

Fig 2B shows 58 carbon sources ranked by the likelihood that cross-feeding evolves for them. Based on our minimal change criterion, cross-feeding of 18 carbon sources can evolve more easily than acetate cross-feeding–it requires fewer reaction changes in producer and consumer. In contrast, cross-feeding of only three carbon sources is easier to evolve than glycerol cross-feeding. In sum, acetate and glycerol are not exceptional in their potential to evolve in cross-feeding.

Thus far, we have used the sum of the ancestor-producer and ancestor-consumer distances as a proxy of the likelihood of the cross-feeding interaction to evolve. By doing so we are inherently assuming that producers and consumers evolve independently from each other. This is strictly not correct, because the producer needs to evolve before the consumer does, otherwise the consumer will lack a carbon source on which to feed. However, because a more complex model (S5 Text) yields essentially identical predictions (see S5 Fig), we will continue to use the sum of metabolic distances below.

Fig 2B uses bars with different colors to distinguish reactions that are activated ('turned on'), inactivated ('turned off'), and that change flux magnitude relative to the ancestor. Just as for acetate and glycerol cross-feeding (Fig 1G–1J), most reactions change their flux quantitatively rather than qualitatively (being turned on or off). Of special interest are those reactions that change flux qualitatively, because it is possible that such flux change is more difficult to achieve genetically, for example because fewer mutations eliminate a gene than modulate its activity [29]. For acetate cross-feeding to emerge, 23 reactions are turned on or off, a number that is lower for five other carbon sources. Likewise, for glycerol-cross feeding to emerge, 20 reactions are turned on or off, a number that is lower in four other carbon sources. Thus, even if the number of reactions turned on or off were the most appropriate measure of metabolic distance, acetate and glycerol cross-feeding would not be exceptional in their metabolic distance to the ancestor. Cross-feeding of other carbon sources requires even fewer qualitative changes than cross-feeding of acetate and glycerol.

The reactions which change flux overlap to some extent across different cross-fed carbon sources. Specifically, out of the 2583 reactions present in *E. coli* model *i*JO1366, 392 (390)

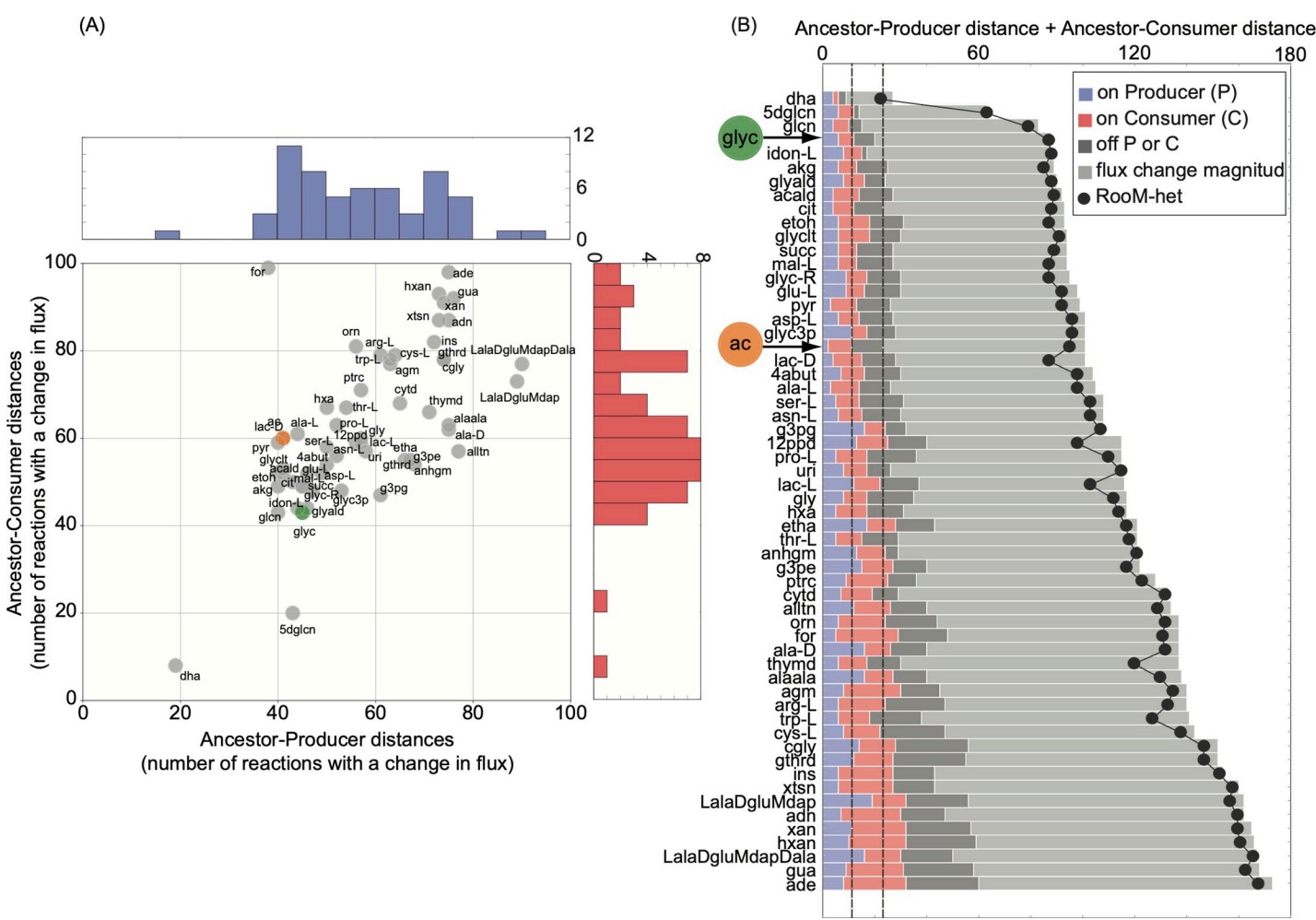

**Fig 2. Metabolic distance between the ancestor and cross-feeding strains.** (A) The ancestor-producer and ancestor-consumer distances (measured as the number of reactions requiring a flux change) obtained when performing RooM are given on the x- and y- axes respectively. Every circle in the plot corresponds to one metabolite with cross-feeding potential. Acetate and glycerol are shown as orange and green circles respectively. The diagonal line is shown as a visual guide. A circle on the line indicates that the same number of reactions needs to change their flux to create the corresponding producer and consumer strain. Blue and red histograms show the distribution of the ancestor-producer and ancestor-consumer distances, respectively. (B) The y-axis shows acronyms for the 58 metabolites that can lead to stable cross-feeding interactions, ranked according to decreasing probability of evolving cross-feeding, as quantified by the total RooM-predicted distance (sum of ancestor-producer and ancestor-consumer distances) shown on the x-axis. Different bar colors indicate the number of reactions classified as turned 'on' in the producer relative to the ancestor (blue), turned 'on' in the consumer relative to the ancestor (red), turned 'off' in either the producer or consumer relative to the ancestor (dark grey), as well as reactions requiring a flux change in either producer or consumer relative to the ancestor ('flux change magnitude', in light grey). Black circles show the total distances obtained when we used RooM-het (explained in the following section) to minimize strain distances.

reactions are required to change their flux to convert the ancestor into a producer (consumer) of at least one of the 58 carbon sources. Nine (four) reactions require a flux change for the evolution of every producer (consumer) (see S2 Text for a list of these reactions). 36 (40) changing reactions are shared among 80% of all producer (consumer) strains. This overlap suggests that the mutations required to create producer and consumer strains of different metabolites may also overlap.

In sum, if metabolic distance is an appropriate proxy for the likelihood that cross feeding evolves, acetate and glycerol cross-feeding are not exceptionally likely to evolve compared to cross-feeding on 56 other carbon sources. This is further supported by our observation that the reactions that require a flux change for the evolution of producers or consumers overlap among different cross-fed carbon sources.

### Can a heterogeneous ancestral population affect the likelihood that cross-feeding emerges?

Thus far we assumed that the ancestral population is homogeneous, i.e., it is composed of phenotypically identical individuals. This is why we modeled it with a single flux distribution predicted through pFBA. However, bacterial populations are often phenotypically heterogeneous. This heterogeneity may arise from genetic differences among individuals in a population, or from noisy gene expression in genetically homogeneous (isogenic) populations. Likewise, in the chemostat experiment that inspired this work, the isogenic ancestral strain may have been phenotypically heterogeneous, or it may have diversified genetically before cross-feeding emerged. Such heterogeneity can affect ecological and evolutionary processes.

In this section we ask how such heterogeneity might affect the evolution of cross-feeding. To this end, we developed a method we call "RooM-het" which allows us to identify the cross-feeding strains that most likely evolve from a heterogeneous ancestral population. As opposed to RooM, RooM-het does not use a single (ancestral) flux distribution as reference. Instead, it identifies two minimally distant flux distributions simultaneously, each fulfilling a different set of constraints (See Fig 3A and 'Methods' for details).

We applied RooM-het to identify producer and consumer strain for each of our 58 carbon sources that can lead to cross feeding. Assuming a heterogeneous ancestral population led to a lower distance to the ancestor in producer and consumer strains. However, the distance reduction was only modest (two changed reactions on average, S6 Fig). As in the previous section, we used the sum of the ancestor-producer and ancestor-consumer distances as a proxy of the likelihood that the different cross-feeding interactions emerge. The black circles in Fig 2B show the total distance to the ancestor as obtained with RooM-het. Taking into account population heterogeneity affects the likelihood of cross-feeding only modestly, changing the rank of acetate (glycerol) cross-feeding from 19-th (4-th) to 18-th (5-th) most likely to evolve. Thus,

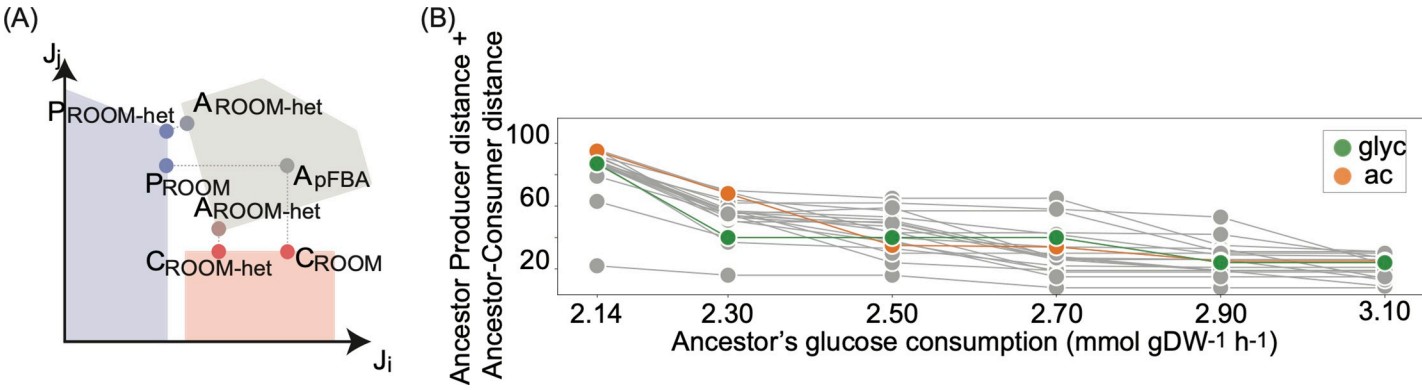

**Fig 3. Impact of a heterogeneous ancestral population on the evolutionary outcome.** (A) Explanation of the RooM and RooM-het methods. The figure shows a hypothetical flux space where the fluxes through reaction $i$ ($J_i$) and $j$ ($J_j$) are shown. Different carbon source consumption and production rates impose different constraints that affect the allowable solution space of ancestor, producer and consumer strains differently. The allowable solution space corresponds to the space of flux distributions that fulfill a set of constraints, here shown as grey, blue and red areas for the ancestor, producer and consumer respectively. To perform RooM we first identified an ancestral flux distribution with pFBA (here represented as the grey circle labeled $A_{pFBA}$). We used this flux distribution as a reference to identify the producer (blue circle labeled $P_{ROOM}$) and consumer (red circle labeled $C_{ROOM}$) flux distributions that would require the minimal number of flux changes. In contrast, RooM-het requires no ancestral reference flux distribution. When using this method to identify a producer flux distribution, the method returns two flux distributions, one that satisfies all the constraints of being a producer, and another that satisfies all the constraints of being an ancestor. The same holds when predicting a consumer flux distribution. In the figure, the resulting producer and consumer flux distributions are shown as blue and red circles labeled $P_{ROOM-het}$ and $C_{ROOM-het}$. The two distinct ancestral flux distributions that result when identifying the producer and consumer distribution are labeled $A_{ROOM-het}$. (B) Sum of the ancestor-producer and ancestor-consumer distances (vertical axis) as a function of the glucose consumed by the ancestral population (horizontal axis). Predictions for acetate and glycerol are shown in orange and green, respectively. Grey circles correspond to predictions for one of the twenty metabolites with a predicted likelihood of being subject to evolved cross-feeding greater than that observed for acetate when either RooM or RooM-het are performed at the minimal glucose consumption rate of 2.14 mmol gDW$^{-1}$ h$^{-1}$.

acetate and glycerol cross-feeding are not especially likely to evolve even when heterogeneous ancestral populations are considered. This result also suggests that our initial results (Fig 1) are insensitive to the ancestral flux distribution used.

## Greater glucose consumption can modify the likelihood of cross-feeding interactions to emerge

Thus far, we assumed that the ancestral strain was maximally efficient in consuming glucose, that is, it consumed the minimal amount of glucose required for persistence in the chemostat at a dilution rate of 0.2 h$^{-1}$. Reducing this efficiency, that is, allowing more than this minimal glucose consumption, may affect metabolism in multiple ways [30]. It may open alternative ways of metabolizing carbon, increase the production of waste products, and in doing so, increase population heterogeneity. Here we explore how such increased glucose consumption may affect the likelihood of observing different cross-feeding interactions. We focus on 20 metabolites, which comprise acetate and the 19 metabolites (including glycerol) whose likelihood to be cross-fed is higher than acetate based on either RooM or RooM-het.

In this analysis, we employed again RooM-het but now we varied the ancestral glucose consumption from the minimum required for growth at 0.2 h$^{-1}$ (2.14 mmol gDW$^{-1}$ h$^{-1}$) to a value of 3.1 mmol gDW$^{-1}$ h$^{-1}$, which corresponds to the glucose consumption required by the gluconate (glcn) producer strain, which has the highest glucose requirement.

As the ancestor consumes increasing amounts of glucose, the metabolic change required for the emergence of producer and consumer strains decreases substantially (S7 Fig). For example, when glucose consumption is minimal (2.14 mmol gDW$^{-1}$ h$^{-1}$) the mean ancestor-producer distance for the twenty metabolites of interest equals 41±7 changed reactions, which reduces to 25±8 reactions when glucose consumption increases by only 7% (2.3 mmol gDW$^{-1}$ h$^{-1}$). This ancestor-producer distance declines to zero for all carbon sources above some threshold of glucose consumption, meaning that the ancestor already produces the cross-fed metabolite without any flux change when it consumes a sufficient amount of glucose. The mean ancestor-consumer distance also decreases substantially (from 44±10 to 29±6 reactions with a 7% increase of glucose consumption), but it generally does not decline to zero. The relative proportion of reactions that are activated, inactivated, or that merely changed flux magnitude relative to the ancestor does not change as glucose consumption increases (S7 Fig).

In sum, ancestor-producer and ancestor-consumer distances become smaller when the ancestor can consume more glucose than minimally necessary to persist in the chemostat. Fig 3A provides a geometric intuition for this observation. Increasing glucose consumption by the ancestral strain results in a larger allowable solution space for this strain (grey area in Fig 3A), because the carbon it consumes can be metabolized in a greater number of alternative (and possibly less efficient) ways. As the allowable solution space for the ancestor increases, the distance to the producer's and consumer's solution space cannot increase as well—it can only remain unchanged or decrease.

The extent to which ancestor-producer and ancestor-consumer distances decrease when the ancestor consumes more glucose depends on the cross-fed metabolite (Fig 3B). This dependency may also affect the ranking of metabolites most likely to be involved in the evolution of cross-feeding as glucose consumption changes. For example, Spearman's correlation coefficient of these ranks varies from a minimum of 0.42 (when changing glucose consumption from 2.3 to 2.5 mmol gDW$^{-1}$ h$^{-1}$) to a maximum of 0.89 (when changing glucose consumption from 2.7 to 2.9 mmol gDW$^{-1}$ h$^{-1}$). Acetate cross-feeding evolution ranks highest among all glucose consumption rates (i.e., it shows the lowest distance to the ancestor) when glucose is consumed at a rate of 2.5 mmol gDW$^{-1}$ h$^{-1}$ (orange circles in Fig 3B). Even then,

however, cross-feeding involving dihydroxyacetone (dha), D-gluconate (glcn), glycolate (glyclt) and glyceraldehyde (glyald) are more likely to evolve. Glycerol ranks highest (fourth) at a glucose consumption rate of 2.3 mmol gDW$^{-1}$ h$^{-1}$(green circles in Fig 3B), where it is outranked by dihydroxyacetone (dha), glyceraldehyde (glyald), and 5-dehydro-D-gluconate (5dglcn).

In sum, even at elevated glucose consumption, cross-feeding of several metabolites is more likely to evolve than cross feeding of acetate and glycerol. However, this likelihood is sensitive to glucose consumption, which shows that interactions between a metabolism and its environment are critical to determine the likelihood that cross-feeding emerges.

## Discussion

Based on our previous predictions that 58 different metabolites can sustain stable communities of cross-feeding *E. coli* bacteria which emerge in a glucose limited chemostat [28], we here identified the minimal amount of metabolic change (numbers of reactions with changed metabolic flux) required for the emergence of each such cross-feeding interaction. We used this amount of change as a proxy for the likelihood of cross-feeding to evolve, with more change implying a lower likelihood of evolution. Regardless of the cross-fed carbon source, the required change was complex. It involved multiple biochemical reactions, metabolic pathways, operons, and regulons. These observations are consistent with a large number of mutations and regulatory changes observed in experimentally identified cross-feeding communities [13,16].

Most importantly, our analysis predicts that the experimentally observed cross-fed metabolites acetate and glycerol are not the most likely to be involved in cross-feeding interactions. Multiple other metabolites can evolve cross feeding through similar or less metabolic change. This prediction is independent of how we quantified the amount of metabolic change (S4 Text and S4 Fig), how we computed the likelihood of cross-feeding to emerge (S5 Text and S5 Fig), or whether we took the heterogeneity of an ancestral population into account (S6 Fig). However, we note that the amount of metabolic change required to evolve cross feeding interactions is sensitive to the amount of glucose consumed by the ancestral population, and the extent of this sensitivity depends on the cross-fed metabolite. Thus, the likelihood to evolve cross-feeding depends not only on the reaction complement of a metabolism, but also on interactions between this metabolism and the environment.

For two reasons, cross-feeding may emerge more easily in chemostats than in batch culture. First, theory shows that for cross-feeding to evolve in batch cultures, the secondary carbon source has to be produced at a high rate [31], which reduces the likelihood that cross-feeding emerges. Second, in chemostats operating at low dilution rates (such as the ones we are considering) mutants with high affinity for the available carbon source are favored and will accumulate in the population [32–34]. Because such mutants consume a lot of the carbon source, they may not metabolize all of it completely, and may thus excrete metabolic by-products (similarly to what occurs in overflow metabolism [35]). In other words, just as for our analysis of excess glucose consumption, producer strains can emerge with little or no metabolic flux changes, which also facilitate the emergence of consumer strains (Fig 3B).

Our analysis has two main limitations. First, we assumed that the most frequently evolving producers and consumers are those requiring the least amount of metabolic change, which we used as a proxy for the smallest amount of genetic change. However, it is well known that the relationship between genetic and phenotypic (metabolic) change is not straightforward. Whereas some DNA mutations may affect only one biochemical reaction, others may affect multiple reactions. What is more, the same amount of phenotypic change in different

individuals may be caused by different numbers or kinds of mutations [15,17,19,36,37]. Although these factors will reduce any association between metabolic change and genetic change, one would expect some statistical association between the two whenever multiple mutations must be responsible for the observed phenotypic change. This is probably the case for cross-feeding, where the minimal metabolic changes affect dozens of reactions in multiple biochemical pathways, modulating them and their regulation–which is driven by multiple regulons and operons–both qualitatively and quantitatively.

Second, we tacitly assumed that genetic change causes the metabolic differences leading to cross-feeding. However, phenotypic plasticity may also be involved, especially for the consumer strain. Once a producer strain has evolved and excreted a new metabolite, other individuals in the population may sense the new metabolite and respond accordingly, possibly through a change in gene regulation that does not require mutations. Such plasticity may be important for yet-to-be-discovered instances of cross-feeding, but we know that it is not solely responsible for experimentally characterized cross-feeding interactions. For example, mutations in the regulatory region of gene *acs*, which expresses the enzyme acetyl CoA synthetase needed for acetate uptake, occurred in all acetate consumer strains that evolved in parallel chemostat experiments [15]. Such parallel evolution indicates that the mutations may be required for the evolution or maintenance of cross-feeding interaction [38]. Similarly, three mutations are required when the acetate consumer strain that evolved in batch experiments is to invade and coexist with the acetate producer strain [16].

There may be several reasons why only acetate and glycerol cross-feeding have been observed experimentally, even though multiple other carbon sources may be just as likely to evolve cross-feeding. First, we cannot exclude that other cross-feeding interactions did evolve but went undetected, because detecting cross-feeding requires extensive genetic and biochemical analysis. In addition, cross-feeding strains must reach a sufficiently high population frequency to become detectable. In previous work [28] we showed that about half of the metabolites we study here would support lower community biomass than acetate cross-feeding. Thus, the frequency of some cross-feeding strains in a population may be low and hard to detect (see S6 Text and S9 Fig). Second, the excretion of some metabolites may have negative effects on growth if the metabolite is toxic or if it changes the pH of the environment in unfavorable ways. Third, we cannot rule out that the consumption of acetate or glycerol might give an advantage to the consumer strain that derives from more than just their role as carbon sources. A precedent for this possibility exists [39]. When a population of yeast cells auxotrophic for lysine encountered lysine limitation, coexistence of the ancestral lysine auxotroph strain and a mutant organosulfur auxotroph repeatedly evolved. The organosulfur auxotroph strain persisted in the population, because it consumed organosulfurs excreted by the ancestral strain. Organosulfur auxotrophy conferred an advantage to the mutant strain, because it recovered the proper nutrient-driven growth regulation that had been impaired in the ancestral strain [39]. Fourth, cross-feeding interactions may be transient. Given sufficient time, a generalist strain that combines maximum glucose uptake with the ability to recover a secondary metabolite could evolve [40,41]. Evolution experiments in environments alternating between pairs of carbon sources showed that such generalists evolve when both carbon sources are metabolized in similar ways, whereas specialists evolved when carbon sources are metabolized differently [42]. Based on this observation, one would expect that a generalist might replace a cross-feeding polymorphism if glucose and the cross-fed metabolite are metabolically similar. Fifth, the evolution of acetate and glycerol cross-feeding in the chemostat experiments may have been facilitated by the initial genotype. The reason is that the ancestral genotype in these experiments harbored regulatory mutations that prevented cells from recovering excreted acetate and overexpressed the glycerol regulon [14,15].

In this contribution, we only studied 58 cross-feeding interactions that can evolve in a single minimal glucose-limited environment. Hundreds or thousands of other cross-feeding interactions can evolve in minimal environments with different primary carbon sources [28], and many more interactions are conceivable in complex environments. To validate such cross-feeding predictions through long term evolution experiments that directly assay for such cross-feeding remains an important task for future work. However, even if only a small fraction of these interactions can be experimentally verified, cross-feeding will emerge as an important source of biodiversity in unstructured and homogeneous environments.

## Methods

### Flux balance analysis (FBA)

Flux balance analysis (FBA) is a computational method to predict metabolic fluxes–the rate at which chemical reactions convert substrates into products–of all reactions in a genome-scale metabolic network [43]. It has been successfully used in many applications, for example to study bacterial growth in different environments [44] or in response to gene deletions [45].

FBA requires information about the stoichiometry of chemical reactions in a metabolic network. It makes two central assumptions. The first is that cells are in a metabolic steady-state. The second is that cells effectively optimize some metabolic property such as biomass production (growth). Additional constraints can be incorporated into the optimization problem that FBA solves, in order to account for the thermodynamic and enzymatic properties of a network's biochemical reaction. The optimization problem that FBA solves can be formalized as a linear programming problem [43,46] in the following way:

$$\text{Max } v_{growth}$$

$$s.t.\ Sv = 0$$

$$l_i \leq v_i \leq u_i$$

Here, $S$ is the stoichiometric matrix, a matrix of size $m{\times}r$ that mathematically describes the stoichiometry of the modeled network's metabolic reactions. The integer $m$ denotes the number of metabolites, and $r$ denotes the number of biochemical reactions in the network. These reactions include all known metabolic reactions that take place in an organism, which are called internal reactions. They also include reactions that represent the exchange (import or export) of metabolites with the external environment. Furthermore, they include a biomass growth reaction, which is a "virtual" reaction that reflects in which proportion biomass precursors are incorporated into the biomass of the modeled organism [20,43,46]. Each entry $S_{ij}$ of the stoichiometric matrix contains the stoichiometric coefficient with which metabolite $i$ participates in reaction $j$. The vector $v$ is a vector (of size $r$) whose entries $v_i$ represent the metabolic flux through reaction $i$ in the network. $v_{growth}$ specifies the flux through the biomass growth reaction. Fluxes through biochemical reactions are restricted by lower and upper bounds that constrain the flux through each reaction in the network. These bounds are given by the variables $l$ and $u$, respectively, which are vectors of size $r$.

### Identification of the flux distribution that characterizes the ancestral strain

Cross-feeding interactions evolve when *E. coli* cells are grown in a glucose-minimal chemostat environment [10]. In this experiment, the ancestral strain, i.e., the strain present at the

beginning of the experiment, is able to grow at a rate equal to the dilution rate of the chemostat (0.2 h$^{-1}$) while consuming the only carbon source present (glucose).

Mirroring these conditions, we used the genome scale metabolic model of *E. coli* iJO1366 [20] and simulated a minimal chemical environment containing glucose as the sole source of carbon. We set glucose consumption to a maximum of 10 mmol gDW$^{-1}$ h$^{-1}$, an arbitrary value based on typical glucose uptake rates in *E. coli* [20,46]. We assumed that ammonium, calcium, chloride, cobalt, copper, iron, magnesium, manganese, molybdate, nickel, oxygen, phosphate, potassium, protons, sodium, sulphate and zinc are present in non-limiting amounts. More-over, we assumed that our (simulated) ancestral strain was able to grow at a dilution rate of 0.2 h$^{-1}$. Then, we used parsimonious Flux Balance Analysis (pFBA) [21] to identify the flux distribution that best describes the ancestral strain.

pFBA is a variation of FBA. It embodies the hypothesis that organisms have evolved the ability to grow at a maximally possible rate but at a minimal cost, for example, in the form of enzyme expression. Its predictions are more accurate than those obtained with traditional FBA [21]. pFBA identifies the flux distribution that satisfies a given growth rate (for example growth at the chemostat dilution rate) while minimizing the sum of all fluxes—a proxy for the total energetic cost of expressing enzymes and transporters. This optimization can be formalized as:

$$\text{Min} \sum |a_i|$$

$$s.t.\ Sa = 0$$

$$l_i \leq a_i \leq u_i$$

$$a_{\text{growth}} = 0.2$$

We performed pFBA in python, using the cobrapy package [47].

Using pFBA we identified a flux distribution (*a* for ancestral) that satisfies the growth rate and glucose consumption constraints while minimizing the total flux. With this flux distribution, the *E. coli* metabolism supports growth at the dilution rate of 0.2 h$^{-1}$, and completely oxidizes 2.1 mmol gDW$^{-1}$ h$^{-1}$ of glucose consumed, excreting carbon dioxide as the sole carbon containing metabolite.

## Regulatory on/off Minimization (RooM)

RooM was originally proposed [48] to study the effects of genetic perturbations on a metabolism. In the present work, we used RooM to identify the evolved flux distributions, utilizing the ancestral flux distribution, obtained with pFBA, as a reference. RooM identifies a flux distribution that fulfills a set of constraints while minimizing the number of reactions whose flux changes relative to a reference flux distribution. It solves a mixed integer linear programming problem that can be formalized as follows:

$$\text{Min} \sum f_i^{internal}$$

$$s.t.\ Se = 0$$

$$f_i \in \{0, 1\}$$

$$l_i \leq e_i \leq u_i$$

$$e_i - f_i(u_i - (a_i + \beta)) \leq a_i + \beta$$

$$e_i - f_i(l_i - (a_i - \beta)) \geq a_i - \beta$$

As in FBA and pFBA, $S$ is the $m \times r$ stoichiometry matrix and $l_i$ and $u_i$ are lower and upper bounds, respectively, that constrain the flux through each reaction in the network according to thermodynamic and capacity constraints. $f_i$ is a binary variable. It takes a value of 1 if reaction $i$ shows a substantial change in flux $e_i$ relative to the reference flux $a_i$ and zero otherwise, where $\beta$ specifies the amount of flux change that is considered substantial. We used an arbitrary value of $\beta = 0.001$ mmol gDW$^{-1}$ h$^{-1}$, as in [48]. However, we note that our observations are not sensitive to this value of beta: Changing $\beta$ by up to 25-fold showed only slight differences in the results (see S8 Fig). In our simulations, the reference flux distribution $a$ corresponds to the ancestral flux distribution obtained with pFBA. We considered only flux changes in internal reactions ($f_i^{internal}$) for optimization with RooM, and performed RooM in python with the optimization solver Gurobi [49].

## RooM-het

We propose a new optimization method that we named RooM-het. We used this method to identify the ancestral and evolved flux distributions, assuming that these distributions may be heterogeneous in ancestral and evolved populations.

In contrast to RooM, no single flux distribution is required as a reference to perform RooM-het. Instead, RooM-het identifies two flux distributions with a minimal distance from each other, where each distribution fulfils a set of constraints. As in RooM, this distance refers to the number of reactions with a significant flux difference between the two distributions. The optimization procedure can be written as:

$$\text{Min} \sum f_i^{internal}$$

$$s.t. \; Sa = 0$$

$$Se = 0$$

$$f \in \{0, 1\}$$

$$l_i^a \leq a_i \leq u_i^a$$

$$l_i^e \leq e_i \leq u_i^e$$

$$e_i - f_i(u_i^a - (a_i + \beta)) \leq a_i + \beta$$

$$e_i - f_i(l_i^a - (a_i - \beta)) \geq a_i - \beta$$

Once again, $S$ corresponds to the $m \times r$ stoichiometry matrix. $f_i$ is a binary variable that takes a value of 1 when the flux through reaction $i$ shows a substantial change between $a_i$ and $e_i$, and zero otherwise. Like in RooM, $\beta$ (with $\beta = 0.001$) specifies the amount of flux change that is considered substantial. $l_i^a$ and $u_i^a$ are lower and upper bounds, respectively, on the fluxes in the ancestral flux distribution. They constrain the flux through each reaction in the network

according to thermodynamic and capacity constraints. Similarly, $l_i^e$ and $u_i^e$ are lower and upper bounds on the fluxes in the evolved flux distributions. Differences in the carbon sources that the various strains consume or produce are introduced by adjusting these bounds.

In contrast to RooM, where the ancestral flux distribution is first calculated with pFBA and then used as reference, with this method the ancestral flux distributions *a* and the evolved flux distribution *e* strains are identified in the same optimization procedure. Repeating the optimization in order to identify different evolved flux distributions may result in the identification of different ancestral flux distributions. This is why this method can account for potential flux heterogeneity in the ancestral population. We solved RooM-het using the optimization solver Gurobi [49].

## Supporting information

**S1 Text. This text file includes a list of acronyms for the reactions requiring a flux change for each of the 58 producer and consumer strains to evolve.** The flux through each reaction is given for the ancestral and evolved strain.
(DOCX)

**S2 Text. Text file listing the acronyms of those reactions requiring a flux change for the evolution of every producer or consumer strain.**
(DOCX)

**S3 Text. Text file explaining how we compared experimentally measured gene expression changes and computationally predicted flux changes.**
(DOCX)

**S4 Text. Two alternative methods to identify flux distributions in cross-feeding strains, and a comparison of their predictions to those of RooM.**
(DOCX)

**S5 Text. A model to estimate the likelihood of cross-feeding to evolve, taking into account that the evolution of producer and consumer strains may not be independent events.**
(DOCX)

**S6 Text. Speculations on still unknown cross-feeding interactions most likely to be found.**
(DOCX)

**S1 File. Mutation analysis.** The file shows all reactions that we predict to require a flux change (columns labeled 'Reaction id') for the evolution of the acetate and glycerol cross-feeding strains. The file also shows the Blattner id and name of all genes associated with a reaction (columns labeled 'Gene id' and 'Gene name' respectively). If a gene was found to be mutated in the chemostat experiments (Kinnersley et al. 2014) we indicate the number of mutations found in the gene (columns labeled 'Mutations (# hits)').
(XLSX)

**S2 File. Experimentally observed data on gene expression changes for the 72 *E. coli* metabolic genes whose expression changed in at least one of the cross-feeding strains.** The table includes the Blattner id for each gene (column B), the gene's expression change in the acetate consumer (column C), in the acetate and glycerol producer (column D), in the two glycerol consumer strains (columns E and F), as well as the reactions associated with each gene (column G).
(XLSX)

**S1 Fig. Metabolic changes required for the evolution of the acetate and glycerol cross-feeding strains.** Like Fig 1, the network corresponds to the central carbon metabolism of *E. coli*. Every orange circle represents a metabolite, and every line a reaction. Thick grey, blue and red lines indicate a non-zero flux in the ancestral, producer and consumer strains respectively. Unlike Fig 1, one can zoom into this figure to see metabolite names, reaction names, and flux values.
(PDF)

**S2 Fig. Comparison of ancestor-consumer distances and B) the sum of ancestor-producer and ancestor-consumer distances when the consumer strains consume (A) only the specific secondary carbon source or (B) both glucose and the secondary carbon source.** The x-axes show the ancestor-consumer and total distances obtained when the consumer strains cannot consume glucose but only the specific secondary carbon source (see also Fig 2A and 2B). The y-axes show the same distances but obtained when the consumer strain consumes 1 mmol gDW$^{-1}$ h$^{-1}$ of glucose and the respective secondary carbon source in amounts that allow growth at 0.2 h$^{-1}$. The ancestor-producer distances used to calculate the total distances shown in (B) are those shown on the x-axis of Fig 2A. Every grey circle represents one of 56 metabolites that can be cross-fed. Acetate and glycerol are shown as orange and green circles, respectively. Even though the ancestor-consumer distances change when the consumer strains consume glucose in addition to their specific secondary carbon source, the main conclusion of this work do not change: Multiple changes are required for any cross-feeding interaction to evolve, and cross-feeding of multiple metabolites may evolve with higher likelihood than acetate and glycerol cross-feeding.
(PDF)

**S3 Fig. Comparison of gene expression data of the cross-feeding strains experimentally evolved in glucose minimal chemostats with computationally predicted flux changes.** The light grey areas in figures (A) to (E) show the number of genes found to be up-regulated, down-regulated, or unchanged in expression for the experimentally observed acetate producer CV103, the acetate consumer CV101, the glycerol producer CV103, and the two glycerol consumer strains CV115 and CV116. The numbers above the grey bars add up to the total number of genes included in the metabolic model of *E. coli i*JO1366 (1367). Green and orange bars show the genes predicted to be up-regulated, down-regulated or unchanged, based on flux changes for reactions associated with these genes, as predicted by (from left to right) RooM, MoMA, minimizing reaction subsets, and RooM-het. Green bars (overlapping the grey area) indicate the number of genes correctly predicted to be up-regulated, down-regulated, or unchanged (true positives). Orange bars indicate the number of genes computationally predicted but not experimentally observed to be up-regulated, down-regulated, and unchanged (false positives). (F) Summarizes the data shown in (A) to (E). For each strain (rows) and each gene category and prediction method (columns), the two numbers separated by a dash indicate the number of true positives and false positives. '*' indicates p<0.05, and '**'p<0.01, based on a Fisher's exact test of the null-hypothesis that the number of genes correctly predicted to be up-regulated, down-regulated, or unchanged can be attributed to chance alone.
(PDF)

**S4 Fig. Comparison of the ancestor-producer, ancestor-consumer and total distances obtained when using different methods to identify flux distributions of cross-feeding strains.** In figures (A) to (C) flux distribution distances predicted by MoMA (on the y-axis) are compared with distances predicted by RooM (on the x-axis). The ancestor-producer distance, the ancestor-consumer distance, and the total distance (sum of ancestor-producer and

ancestor-consumer distances) are shown in panels (A) to (C), respectively. Every grey circle represents one of 56 metabolites that can be cross-fed. Acetate and glycerol are shown as orange and green circles. Panels (D) to (F) are analogous to (A) to (C), but their y-axes show the distances predicted when minimizing the number of co-regulated reaction subsets that change expression.
(PDF)

**S5 Fig. Estimates of likelihood of cross-feeding to evolve considering the evolution of producer and consumer strains are not independent events.** (A) The cumulative probability for the evolution of cross-feeding interactions as calculated with Eq (2) from S5 Text is plotted against time (in generations). Every grey line corresponds to a prediction for a different metabolite subject to cross-feeding. Predictions for acetate and glycerol are shown in orange and green, respectively. (B) Comparison of two proxies of the likelihood that cross-feeding evolves. The x-axis shows the sum of the producer-ancestor and consumer-ancestor distances, which is the proxy used in the main text. The y-axis shows the cumulative probability of cross-feeding to evolve after 100 generations, according to the model from S5 Text, which takes into consideration that the evolution of producer and consumer may not be independent events. Every grey circle represents a prediction for a different metabolite subject to cross-feeding. Orange and green circles correspond to predictions for acetate and glycerol, respectively. The two proxies for the likelihood to evolve cross-feeding are highly correlated (Spearman's r = 0.99, P = 9.7e-74, n = 58).
(PDF)

**S6 Fig. RooM-het and RooM distances comparison.** (A) Ancestor-producer distances predicted with RooM and RooM-het are shown on the x and y-axes respectively. Every circle corresponds to one produced metabolite. The diagonal line indicates equal distances. (B) As in (A) but for predicted ancestor-consumer distances and for consumed metabolites.
(PDF)

**S7 Fig. The x-axes in all panels show the maximal amount of glucose consumed by the ancestor.** As a function of this quantity, (A) and (B) show the predicted distance of the ancestor to the producer and consumer, respectively. Among all reactions with a change in flux, (C), (E) and (G) show the fraction of reactions that are turned 'on', 'off' and that change the flux quantitatively in the producer relative to the ancestor, respectively; (D), (F) and (H) are analogous to (C), (E) and (G), but for the reactions that require a flux change in the consumer relative to the ancestor. Each set of six grey circles connected by a grey line corresponds to simulation data for one of the twenty metabolites with a predicted likelihood of being subject to evolve cross-feeding greater than that for acetate when either RooM or RooM-het are performed at the minimal glucose consumption rate of 2.14 mmol gDW$^{-1}$ h$^{-1}$. Predictions for acetate and glycerol are shown as orange and green circles, respectively.
(PDF)

**S8 Fig. The figure shows the ancestor-producer and ancestor-consumer distances (y-axis in the upper and lower panel respectively) predicted by RooM for all 58 cross-feeding strains, using three different values of the parameter beta, which is used in RooM to specify the amount of flux change that is considered substantial (increasing distance in one unit).** The data shows that changing beta from its default value (0.001 mmol gDW$^{-1}$ h$^{-1}$, green circles) to a fifth of this value (0.0002 mmol gDW$^{-1}$ h$^{-1}$, yellow circles), or to five times this value (0.005 mmol gDW$^{-1}$ h$^{-1}$, grey circles) has very little effect on the predicted distances.
(PDF)

**S9 Fig. The horizontal axis shows the total metabolic distance to the ancestor found with RooM (as in Fig 2B).** On the vertical axis, colored circles next to each metabolite's acronym indicate the product of maximal metabolite production and biomass yield, where the same color code as in Fig 2 from [28] is used. Specifically, community biomass increases from yellow to green to blue. The figure shows that cross-feeding interactions whose evolution requires few metabolic changes (i.e., low ancestor-producer plus ancestor-consumer distances) usually result in high community biomass (blue circles). Pink circles indicate carbon sources that *E. coli* can excrete when growing in glucose minimal medium [50].
(PDF)

## Author Contributions

**Conceptualization:** Magdalena San Roman, Andreas Wagner.

**Data curation:** Magdalena San Roman, Andreas Wagner.

**Formal analysis:** Magdalena San Roman, Andreas Wagner.

**Funding acquisition:** Andreas Wagner.

**Investigation:** Magdalena San Roman, Andreas Wagner.

**Methodology:** Magdalena San Roman, Andreas Wagner.

**Project administration:** Magdalena San Roman, Andreas Wagner.

**Resources:** Magdalena San Roman.

**Software:** Magdalena San Roman.

**Supervision:** Andreas Wagner.

**Validation:** Magdalena San Roman, Andreas Wagner.

**Visualization:** Magdalena San Roman, Andreas Wagner.

**Writing – original draft:** Magdalena San Roman, Andreas Wagner.

**Writing – review & editing:** Magdalena San Roman, Andreas Wagner.

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
