## [Decision Letter · Decision Letter 0]

15 May 2020

Dear Mrs San roman,

Thank you very much for submitting your manuscript "Acetate and glycerol are not uniquely suited for the evolution of cross-feeding in E. coli" for consideration at PLOS Computational Biology. As with all papers reviewed by the journal, your manuscript was reviewed by members of the editorial board and by several independent reviewers. The reviewers appreciated the attention to an important topic. Based on the reviews, we are likely to accept this manuscript for publication, providing that you modify the manuscript according to the review recommendations.

Sincerely,

Kiran Raosaheb Patil, Ph.D.

Associate Editor

PLOS Computational Biology

Alice McHardy

Deputy Editor

PLOS Computational Biology

[LINK]

Reviewer's Responses to Questions

**Comments to the Authors:**

Reviewer #1: The authors use flux balance analysis to explore which cross-feeding interactions are most likely to evolve in E. coli growing in a minimal medium containing glucose as the sole carbon source. They show that - according to their models and assumptions - the two that have been found to evolve in previously published experiments, glycerol and acetate, do not seem to be the two that are most likely to evolve. They determine this by calculating the number of metabolic changes (on/off/quantitative change) that are needed for each metabolite to be produced and consumed by mutants. They find that there are other metabolites that require fewer changes to the ancestor compared to the two that were found. Assuming parsimony, those should be more likely to evolve.

I found the analysis to be very thorough and convincing. The assumptions of the approach and its short-comings are very clearly stated, and seem reasonable to me. I find it hard to imagine how to do this otherwise. The paper is also very well written and a pleasure to read, with many interesting findings.

Regarding the conclusion: the authors hypothesise that cross-feeding on many other metabolites may have evolved, but has just gone undetected. You write: “In previous work [27] we showed that about half of the metabolites we study here would support lower community biomass than acetate cross-feeding. Thus, the frequencies of some cross-feeding strains in a population may be low and hard to detect”. These short sentences seem crucial to answering the question. If this is really something that could explain the findings, then I think it merits additional explanation. It was not clear to me why this would be the case. A further point that merits discussion is that each metabolite may have different effects on growth, if, for example they change the pH of the environment in an unfavourable way to the ancestor.

The other thing that would be nice to see is a bit more of a detailed analysis of how these results correspond to what has been seen in the genomic analysis (mutations) of evolved cross-feeding E. coli populations from previous work. The authors state that there is some overlap in the results (“Multiple repeatedly mutated genes [14] were directly involved in glycolysis and gluconeogenesis, the TCA cycle, and transport, which are three of the subsystems where we also observe most of the reactions changes”), and then focus on transcriptomic data. It’s true that the transcriptomic data is a better place to start, but it’s only looking at 72 genes, and there might be something interesting looking at mutations (perhaps in a supplementary figure). If some correspondence is found (even though one would not necessarily expect a lot of overlap, for reasons that the authors already state), it would provide support for the method.

Minor comments:

- You mention that parsimonious FBA is not the standard. Can you say (perhaps in the discussion) what would happen if you used regular FBA? Is parsimonious more conservative? Do you expect to see the same patterns?

- Fig. 1: I find the very small text on the networks a bit annoying. Do you expect the reader to zoom in to read them? I would just remove the text and make the orange circles larger. The point after all is to show large changes.

- Caption of Fig. 2: RooM-het appears but is only explained later in the text. Maybe just warn the reader.

- L. 307: “cross-feeding of 18 metabolites is easier to evolve” than what? I was confused how these 18 compare to the 7 listed in the previous paragraph.

- L. 328: “acetate and glycerol cross-feeding would not be exceptional in their metabolic distance to the ancestor”. No, but they are on the low end, so in the set that is more likely to evolve. Possibly one additional (yet unknown) factor may explain why they preferentially evolve.

- L. 336ff: “This overlap suggests that the mutations required to create producer and consumer strains of different metabolites may also overlap.” To what extent do glycerol and acetate overlap? Do they actually both evolve in the same experiments and if so, might that be explained by a particularly high overlap?

- L. 424: “it is not surprising that the sum of the ancestor-producer and ancestor-consumer distances decreases as glucose consumption increases.” I found this surprising. It would be good to spell out the intuition of why this may be the case. I believe it came later in the discussion, but would be useful to explain it here.

- L. 430: “Acetate ranks highest (fifth)” I was confused how fifth was highest if you are testing 20 metabolites

- L. 486ff: “we tacitly assumed that genetic change causes the metabolic differences leading to cross-feeding. However, phenotypic plasticity may also be involved, especially for the consumer strain”. Genetic change can also occur due to drift or be due to selection for non-metabolic traits such as motility.

- Some small typos here and there (s’s where they shouldn’t be, one figure label says figure_s5.pdf)

Reviewer #2: San Roman and Wagner asked why E. coli in low glucose evolved cross-feeding interactions mediated by acetate and glycerol (and not other metabolites). One possibility is that other metabolites are released and consumed, but these interactions have not been detected. To address this possibility, they used metabolic modelling to estimate the likelihood of evolving cross-feeding interactions by how small the total number of metabolic changes required in the ancestor to give rise to a producer strain and a consumer strain. They found that among the 50+ metabolites predicted to be released (including acetate and glycerol), acetate and glycerol are not really exceptional, and that these cross-feeding interactions would require changes in many metabolic reactions (twenty or so reactions need to be turned on or off). Consistently, previous published experimental work showed that repeatedly mutated genes are involved in three of the subsystems predicted to be important. Their model further predicted that although phenotypic heterogeneity only slightly helped the evolution of cross-feeding, consuming glucose more than what was required for growth in chemostats would significantly facilitate the evolution of cross-feeding. The paper proposes tantalizing hypotheses, and is easy to follow.

Main comments:

1. Why is there such a nice positive correlation between ancestor-producer distance and ancestor-consumer distance?

2. You listed two limitations of FBA modelling in your discussions, whereas in reality, there could be more. For example, your model assumed that all excessive metabolites are released whereas in reality, transporters of metabolites might not be there to facilitate release.

3. A related point of “not detecting cross-feeding” is that the frequency-dependent fitness advantage of consumer over producer is very important for the consumer to rise to a sufficiently high frequency to be detected. As an example, see our preprint https://www.biorxiv.org/content/10.1101/498543v2

**Have all data underlying the figures and results presented in the manuscript been provided?**

Reviewer #1: Yes

Reviewer #2: None

PLOS authors have the option to publish the peer review history of their article (what does this mean?). If published, this will include your full peer review and any attached files.

Reviewer #1: No

Reviewer #2: Yes: Wenying Shou
---

## [Decision Letter · Decision Letter 1]

31 Aug 2020

Dear Mrs San roman,

Thank you very much for submitting your manuscript "Acetate and glycerol are not uniquely suited for the evolution of cross-feeding in E. coli" for consideration at PLOS Computational Biology. As with all papers reviewed by the journal, your manuscript was reviewed by members of the editorial board and by two independent reviewers. Based on the reviews, we are likely to accept this manuscript for publication. We would like you to consider modifying the discussion slightly based on the comment by Reviewer-1.

Sincerely,

Kiran Raosaheb Patil, Ph.D.

Associate Editor

PLOS Computational Biology

Alice McHardy

Deputy Editor

PLOS Computational Biology

[LINK]

Reviewer's Responses to Questions

**Comments to the Authors:**

Reviewer #1: I would like to thank the authors for addressing my questions so thoroughly. I especially appreciate the additional analysis of the experimental data on mutations (now supplementary text S3 and S1 file). I also find that supplementary text S6 adds a satisfying explanation to the existing results. I have no further recommendations for improvement.

Reviewer #2: "A related point of “not detecting cross-feeding” is that the frequency-dependent fitness

advantage of consumer over producer is very important for the consumer to rise to a

sufficiently high frequency to be detected. As an example, see our preprint

https://www.biorxiv.org/content/10.1101/498543v2

We fully agree that a frequency-dependent advantage explains the increase of consumer

abundance in the chemostat, and have already shown this in our previous work (San Roman

and Wagner 2018). In this work, we had simulated the dynamics of a community that initially

harbored only a producer strain. At a later time point we introduced the consumer strain at a

very low abundance, to simulate its evolution. We found that the consumer strain initially

increases its abundance due to a frequency-dependent effect. Specifically, the secondary

carbon source excreted by the producer strain is at first consumed (and needs to be ‘shared’)

only by a few consumer cells. This results in a high growth rate of the consumer strain, and

increased consumer abundance. As the number of consumer cells increases, the available

secondary carbon source has to be shared between more cells, reducing the consumer’s

growth rate. Eventually, a steady-state is reached where both producer and consumer strains

are abundant."

The authors misunderstood me. What I meant is that the fitness advantage part is not a given (no law states that consumers must gain an advantage over producers). This lack of advantage could contribute to "not detecting cross-feeding".

**Have all data underlying the figures and results presented in the manuscript been provided?**

Reviewer #1: Yes

Reviewer #2: Yes

PLOS authors have the option to publish the peer review history of their article (what does this mean?). If published, this will include your full peer review and any attached files.

Reviewer #1: **Yes: **Sara Mitri

Reviewer #2: **Yes: **Wenying Shou
---

## [Editor Report · Decision Letter 2]

10 Oct 2020

Dear Mrs San roman,

We are pleased to inform you that your manuscript 'Acetate and glycerol are not uniquely suited for the evolution of cross-feeding in E. coli' has been provisionally accepted for publication in PLOS Computational Biology.

Best regards,

Kiran Raosaheb Patil, Ph.D.

Associate Editor

PLOS Computational Biology

Alice McHardy

Deputy Editor

PLOS Computational Biology

---

## [Editor Report · Acceptance letter]

4 Nov 2020

PCOMPBIOL-D-20-00163R2 

Acetate and glycerol are not uniquely suited for the evolution of cross-feeding in E. coli

Dear Dr San roman,

I am pleased to inform you that your manuscript has been formally accepted for publication in PLOS Computational Biology. Your manuscript is now with our production department and you will be notified of the publication date in due course.

With kind regards,

Nicola Davies
